# Ensuring the Efficacious Iron Fortification of Foods: A Tale of Two Barriers

**DOI:** 10.3390/nu14081609

**Published:** 2022-04-12

**Authors:** Richard F. Hurrell

**Affiliations:** Institute of Food, Nutrition and Health, ETH Zurich, Schmelzbergstrasse 7, CH 8092 Zurich, Switzerland; richard.hurrell@hest.ethz.ch

**Keywords:** iron fortification, iron status, iron efficacy, infection, inflammation

## Abstract

Iron fortification of foods has always been a challenge. This is because iron fortification compounds vary widely in relative absorption; because many foods undergo unacceptable changes in color or flavor from the addition of iron; and because many of the iron-fortified foods contain potent inhibitors of iron absorption. These technical barriers have largely been overcome, and efficacious iron-fortified foods, that maintain or improve the iron status of women or children in long-term feeding studies, can be designed. Commercially fortified infant foods are efficacious, and other commercial iron-fortified foods targeted at women and children will provide a useful amount of iron provided the fortification level is adjusted according to the relative absorption of the iron compound. Technologies for the large-scale fortification of wheat and maize flour are also well established, and iron fortification of rice, using the recently developed extruded premix technique, is showing great promise. However, some important knowledge gaps still remain, and further research and development is needed in relation to iron (and iodine)-fortified salt and iron-fortified liquid milk. The usefulness of less-soluble iron compounds, such as ferrous fumarate, to fortify foods for infants and young children in low- and middle-income countries (LMICs) also needs further investigation. A more formidable barrier to efficacious iron-fortified food has been reported in recent years. This is the infection-initiated inflammation barrier, which inhibits iron absorption in response to infection. This barrier is particularly important in LMICs where infections such as malaria and HIV are widespread, and gastrointestinal infections are common due to poor quality water supplies and sanitation. Another source of inflammation in such countries is the high prevalence of obesity in women. Most countries in sub-Saharan Africa have high inflammation which not only decreases the efficacy of iron-fortified and iron-biofortified foods but complicates the monitoring of large-scale iron fortification programs. This is because iron deficiency anemia cannot be differentiated from the more prominent anemia of inflammation and because inflammation confounds the measurement of iron status. There is an urgent need to better quantify the impact of inflammation on the efficacy of iron-fortified foods. However, at present, in LMICs with high inflammation exposure, infection control, cleaner water, improved sanitation, and a decrease in obesity prevalence will undoubtedly have a greater impact on iron status and anemia than the iron fortification of foods.

## 1. Introduction

Iron-fortified foods are designed to maintain or improve the iron status of individuals or populations whose dietary iron intake is too low to cover their iron requirements. The population groups most often lacking in iron, and those most often targeted with iron-fortified foods, are women of reproductive age, adolescents, infants, and young children. The higher iron requirement for young women is needed to cover their monthly menstrual blood losses, as well as to provide for an increased iron need for past and future pregnancies. Infants, young children, and adolescents require more iron for the expansion of their blood supply during periods of growth. Iron fortification of staple foods, condiments, and foods for infants and young children is often the chosen strategy to increase dietary iron intake and prevent iron deficiency (ID) and iron deficiency anemia (IDA) and thus prevent the negative health consequences associated with these conditions [1].

The first iron-fortified foods appeared in the 1940s with the enrichment of flour and bread in the United States and the United Kingdom [2] and by the 1980s, there was much interest in the addition of iron to other foods including wheat and maize products, rice, breakfast cereals, infant formulas and complementary foods, milk, soft drinks, salt, and other condiments [3].

Nowadays, food can be iron fortified either by the post-harvest addition of iron compounds during food processing or by the biofortification of staple cereals and legumes by plant breeding or genetic engineering. Commonly consumed iron-fortified foods can be divided into five different categories. The largest category includes the staple foods that form part of large-scale national iron fortification programs. These foods include wheat flour, maize flour, rice grains, common salt, and liquid or powdered milk. National iron fortification programs are often mandated and increase the iron intake of the entire population, including adult men and post-menopausal women who would already be expected to consume enough iron from their regular diet to cover their requirements. The second category is the market-driven iron fortification of infant formulas and complementary foods. These commercial foods are consumed by infants and young children primarily from high-income countries but also from the most affluent sectors of low- and middle-income countries (LMICs). The third category is the food products provided by aid agencies for young children living in LMICs. This category includes complementary foods, micronutrient powders (MNPs), and lipid-based micronutrient supplements. The fourth category contains foods to which iron has been voluntarily added by the food industry and targeted primarily at women, adolescents, and children. This type of fortification is often described as market-driven fortification and includes breakfast cereals, chocolate drink powders, and bouillon cubes. These foods aim to provide a useful amount of bioavailable iron, in contrast to iron-fortified infant foods and staple foods in national programs which are specifically formulated to provide enough iron to fill the gap between the current iron intake of the at-risk population groups and their iron requirement. It should be noted that most of these foods are simultaneously fortified with a range of other micronutrients, in addition to iron, and that their nutrient composition is regulated by national and international standards.

Iron biofortification of cereal staples and roots, by conventional breeding or genetic engineering, is the fifth category and it targets ID in low-income households in remote areas where there is little or no access to manufactured foods [4]. However, because of insufficient genetic variability in the iron levels in the germplasms of wheat, rice, and maize, iron biofortification of cereal staples by plant breeding has not been able to reach the iron levels that are needed for efficacious post-harvest fortification. The most successful iron biofortification programs until now [5] have been with pearl millet, and with legumes, such as common beans, chickpeas, and lentils, where it has been estimated that the higher iron content in these biofortified foods could provide up to 25% of the iron requirement of young women. Biofortification by plant breeding, like market-driven fortification, can thus provide a useful amount of bioavailable iron and, if biofortified foods were part of a national program, they would be regularly consumed by the whole population. However, animal tissue foods or post-harvest iron-fortified foods would still be needed if ID is to be prevented or eliminated in such populations. Genetic engineering procedures could increase the iron content of biofortified staples up to a level where they (alone) would be able to prevent ID.

Iron has proven the most difficult micronutrient to add to foods and to demonstrate an improvement in nutrient status, and this review will focus on the technical barriers encountered when adding iron to manufactured foods and the difficulties encountered when designing an efficacious iron-fortified food that will improve the iron status in women and/or children. The first problem to overcome was that iron compounds vary widely in the extent to which they are absorbed, and they vary in their potential, or not, to cause unacceptable changes in the color and flavor of the food to which they are added. A second challenge has been that common food fortification vehicles, such as cereals foods and milk, contain potent inhibitors of iron absorption. These inhibitory food components were first identified and strategies were then developed to overcome their inhibition and optimize iron absorption from the iron-fortified foods. This enabled iron-fortified foods to be designed with an amount of iron that, when based on estimated consumption, would provide an iron intake that would fill the gap between current iron intake and iron requirements; and then demonstrate the efficacy of these foods in long-term feeding studies in women and children.

However, even though iron-fortified foods have been demonstrated many times to improve the iron status in women and children in well-controlled feeding studies, there are still doubts that large-scale iron fortification programs at the national level are an effective strategy to improve or maintain iron status in all populations. This is in contrast to the very successful food fortification programs with other commonly deficient micronutrients such as iodine, folic acid, vitamin A, and vitamin D. One reason for this difference could be that deficiencies in these micronutrients result in distinct physical changes in an individual’s appearance and are easier to monitor than the deficiency symptoms with ID. Another reason, however, is that the widespread inflammatory disorders and infections, that are common in many LMICs, can inhibit iron absorption and can render the iron status biomarkers unreliable. This makes iron fortification programs in such countries potentially less efficacious and makes it much more difficult for them to demonstrate an improvement in iron status.

The inflammation/infection barrier and the technical barriers are specific to iron, and both must be overcome in order to ensure effective iron fortification in all countries. Ineffective monitoring of large-scale programs is a third barrier that could decrease the performance not only of iron fortification but of all micronutrient fortification programs. While there are many actions needed to ensure the effective monitoring and evaluation of all large-scale micronutrient fortification programs [6], poor regulatory monitoring and enforcement has recently been cited as the major issue of concern. [7].

Most iron fortification research to date has focused on overcoming the technical barrier to manufacturing efficacious iron-fortified foods [8]. This barrier has now largely been overcome with respect to those iron-fortified staple foods that form the basis of national fortification programs (category 1), and all commercially manufactured, iron-fortified foods formulated for infants and young children (category 2) would be expected to be efficacious. Nevertheless, while fortification technologies for wheat flour, maize flour, rice, salt, and milk have all been reported to improve the iron status in women and children, some important knowledge gaps still remain and further research and development is needed particularly in relation to iron-fortified salt, rice, and liquid milk. Similarly, low-cost foods provided by aid agencies for young children in LMICs (category 3) still pose some challenges to guaranteeing adequate iron absorption.

Commercial foods that are voluntarily fortified with iron (category 4) aim to provide a useful amount of bioavailable iron as opposed to preventing ID (categories 1–3). National/international food regulations provide thresholds for the iron fortification of commercial foods for the general population (>3 years) and food companies must provide >15% of the estimated iron requirement per serving [9]. This amount could be considered “useful” provided that it is adjusted based on the relative absorption of the iron fortification compound added.

The ability of inflammation and infections to provoke a hepcidin-induced restriction in iron recycling and a decrease in dietary iron absorption is a relatively recent observation [10,11]. Nevertheless, this inflammation/infection barrier would explain the difficulties observed in demonstrating the efficacy of iron-fortified foods that are consumed by women or children living in LMICs where infections and inflammation are widespread [12]. Safety may also be of concern when malaria is endemic and when hygiene is poor. MNPs, used for in-home fortification of cereal porridges, have been reported to increase fecal pathogens in environments with poor hygiene [13], resulting in bloody stools [14], and potentially exacerbating malaria in children [15]. So far, little or no research has been made on how to improve the efficacy and safety of iron-fortified foods in LMICs where widespread inflammation and infections are common, and it is unclear whether iron-fortified foods can usefully improve the iron status of women and children living in such areas.

This review will first describe the research that was needed to overcome most of the technical barriers and design efficacious iron-fortified foods. The current fortification practices for staple foods, and foods provided by aid agencies for children in LMICs, will then be discussed, highlighting the gaps in our knowledge that need further research. Finally, the possibility of overcoming the inflammation/infection barrier will be discussed, asking whether and how we can provide safe and efficacious iron-fortified foods in LMICs where inflammation and infection are widespread.

## 2. Overcoming the Technical Barrier to Efficacious Iron Fortified Foods

### 2.1. Selecting an Iron Fortification Compound

It has long been known that freely water-soluble iron compounds are better absorbed than poorly soluble iron compounds but often cause color and/or flavor changes to sensitive foods; and that poorly soluble compounds cause no or few sensory changes but are less well absorbed. The iron compound selected for a specific food vehicle is usually the most bioavailable compound that causes no sensory changes at an acceptable cost [16]. According to the WHO [17], the recommended iron compounds for fortification of most foods are, in order of preference, ferrous sulfate, ferrous fumarate, encapsulated ferrous sulfate or fumarate, electrolytic iron (at double the iron amount as ferrous sulfate), and ferric pyrophosphate (at double the amount of ferrous sulfate). Iron compounds used for food fortification were reviewed most recently by Hurrell [18].

#### Relative Bioavailability (RBV) versus Sensory Changes

Early rat studies measured the relative bioavailability (RBV) of a large number of different iron compounds relative to ferrous sulfate (RBV = 100) by comparing their ability to regenerate hemoglobin in iron-deficient rats [19]. RBV in rats varied from 6–120 and iron compounds could be divided into three groups based on their solubility, RBV, and potential for sensory changes. The three groups were those that were readily water soluble; those that were poorly water soluble but soluble in dilute acids, such as the gastric juice; and those that were partially soluble in dilute acid [20]. Iron compounds that were water soluble, or completely soluble in the gastric juice, had an RBV close to 100, whereas those which were only partially soluble in dilute acid had a low and variable RBV depending on the extent to which the iron compound dissolved in the gastric juice during digestion. The water-soluble compounds, however, were the most likely to cause sensory changes in sensitive foods, and the compounds only partially soluble in dilute acid were the least likely. There were also a small number of compounds that were water insoluble and caused fewer or no sensory changes but had an RBV equivalent to ferrous sulfate indicating that they dissolved in the gastric juice during digestion. Ferrous fumarate and ferrous succinate were two such compounds.

RBV obtained from hemoglobin repletion studies in rats turned out to be a good proxy for the relative absorption of iron compounds as measured with radio or stable iron isotope studies in humans [21], and human iron absorption studies gradually consolidated the findings from the rat studies. Hurrell et al. [22] investigated the suitability of different iron compounds to fortify commercial, cereal-based complementary foods and reported that ferrous fumarate and ferrous succinate had an RBV equivalent to ferrous sulfate in humans and ferric pyrophosphate (FePP) an RBV of 40. Ferrous fumarate, which is poorly water soluble, and causes far less sensory changes than ferrous sulfate, quickly became a popular iron fortification compound for commercial cereal-based complementary foods, and encapsulated ferrous fumarate later became the iron compound of choice for MNPs. FePP is water insoluble, but only partially soluble in dilute acid. It causes few if any sensory changes to foods and is now widely used to fortify cereal-based complementary foods that are sensitive to color changes, chocolate drink powders, bouillon cubes, and extruded fortified rice grains [16,18].

Iron compounds, such as FePP, that are only partially soluble in dilute acid, dissolve in the gastric juice to a different extent depending on the meal composition, and absorption relative to ferrous sulfate (RBV) varies with meal composition. In human isotope studies, the RBV of FePP varied from 15% in fortified rice [23] to 75% in a reconstituted chocolate drink powder [24]. Iron absorption from FePP is also less efficiently upregulated than iron absorption from ferrous sulfate resulting in lower RBV values for FePP in subjects with ID than in subjects with an adequate iron status [23]. A lower RBV for ferrous fumarate (RBV = 30) has also been reported in Bangladeshi infants [25] and Mexican children [26] (RBV = 30), in contrast to a similar absorption from ferrous fumarate and ferrous sulfate in Mexican infants, children, and women of normal iron status [27].

It is possible that ID children cannot upregulate iron absorption from ferrous fumarate as well as they can from ferrous sulfate, perhaps due to an incomplete dissolution of ferrous fumarate during digestion in less well-nourished children. This explanation fits with the observations made with FePP [23] and with the results from a recent iron absorption study in which bread fortified with ferrous sulfate or ferrous fumarate was fed to iron-replete Senegalese children (2–6 y) and their mothers [28]. While the RBV of fumarate added to bread was 91 in the mothers, it was only 64 in the children. The authors suggested that the lower RBV of ferrous fumarate in the iron-replete children was due to insufficient gastric acid secretion decreasing the solubility of ferrous fumarate during gastric digestion. Ferrous fumarate is widely used to fortify low-cost blended complementary foods, such as corn soy blend in the US food aid program [29], and encapsulated ferrous fumarate is the compound of choice for micronutrient powders used for home fortification. The possibility that fractional iron absorption by infants and young children from this compound is lower than previously believed is a cause for concern and needs further evaluation.

The use of elemental iron powders to fortify foods is also an issue. Iron powders are the least expensive iron fortification compound and cause few if any sensory changes to the food vehicle. Wheat flour, the first vehicle used for iron fortification in the United States, was fortified with elemental iron powder to prevent rancidity which developed with ferrous sulfate and many other iron compounds, because of the iron-catalyzed oxidation of wheat lipids. Elemental iron powders then became the iron compound of choice in the United States for the fortification of breakfast cereals and cereal-based complementary foods.

However, like FePP, elemental iron powders are only partially soluble in the gastric juice during digestion and have given low and variable RBV values in rat studies [30]. The commercially available iron powders are manufactured by five different processes, with major differences both between and within powder types with respect to size distribution, surface area, dissolution rate, and absorption from food [31]. Rat RBV values indicated that electrolytic iron, carbonyl iron, and H-reduced are about 50% as well absorbed as ferrous sulfate, with iron absorption from CO-reduced iron and atomized reduced iron being unacceptably low [32]. Few human studies have been made with elemental iron powders due to the expense and the difficulty in making isotopically labeled compounds that have the same physical characteristics as the commercial powder.

As it is not possible to control which powder has been added to foods, an expert group recommended that only one well-characterized powder should be used for food fortification, and they selected electrolytic iron [30]. Additional support for that choice came from a radioisotope absorption study in human subjects [21], plasma absorption curves in humans [33], and efficacy studies in young children fed an infant cereal [34] and in women fed wheat flour snacks [35]. Other elemental iron powders still continue to be used in some well-established large-scale programs, such as in the United States, but not in the newly established large-scale programs.

### 2.2. Inhibitors and Enhancers of Iron Absorption

Rat studies are unsuitable for measuring food components that inhibit or enhance iron absorption in humans [36], and this led to some confusion prior to the development of radioisotope and stable isotope absorption studies in human volunteers. The isotope studies have now clearly shown that the main enhancers of dietary iron absorption in humans are ascorbic acid in fruits and vegetables, and partially digested muscle proteins, although no single enhancing component of muscle tissue has yet been identified. Phytic acid in cereal and legume foods is the most important inhibitor of iron absorption with milk and legume proteins, calcium in dairy products, and polyphenol compounds in fruits and vegetables [37] also potential inhibitors of fortified iron.

#### 2.2.1. Ascorbic Acid

Ascorbic acid is the only food component that is routinely added to iron-fortified foods to enhance iron absorption. It can enhance several-fold the absorption of native iron and all iron fortification compounds, except the iron chelates, and it can overcome to a large extent the inhibitory effects of phytic acid, calcium, milk and legume proteins, and phenolic compounds [8]. It acts as a reducing agent and iron-solubilizing ligand in the stomach and duodenum, converting ferric to ferrous iron which less readily forms insoluble hydroxides and less strongly combines with the inhibitory food components. Increasing amounts of ascorbic acid progressively increase iron absorption until a plateau is reached [38]. An ascorbic acid molar ratio of 2:1 is sufficient to increase iron absorption from 2–12 times in most foods, except for high phytate cereal and soy products, where a ratio of 4:1 is needed [8,39]. These molar ratios are recommended by the WHO [17]. The disadvantage of ascorbic acid is that it is sensitive to heat and is readily degraded by oxidation during cooking procedures and during storage at high ambient temperatures in oxygen and humidity permeable packages. More sophisticated packaging will mitigate these losses but will add cost. Ascorbyl palmitate has been reported to be thermostable [40] and may be useful for some applications.

#### 2.2.2. Iron Chelates

Another strategy to enhance the absorption of fortification iron from food vehicles or meals rich in phytic acid and other iron absorption inhibitors is to fortify with the iron chelates NaFeEDTA or ferrous bisglycinate. Iron from NaFeEDTA is 2–3 times better absorbed from high phytate meals than ferrous sulphate [41] and is more efficacious in cereal flours [42]. Although it may cause adverse color changes in sensitive foods, it does not promote fat oxidation in stored cereal flours or precipitate peptides in soy and fish sauces. It is the WHO [43] compound of choice for high phytate cereal flours and in recent years has also been added, together with ferrous fumarate, to low-cost complementary foods such as corn soy blend. This is an important development since the ascorbic acid added to the blend is likely to be degraded during the necessary cooking step.

Ferrous bisglycinate overcomes phytic acid inhibition to a similar extent as NaFeEDTA; however, it can promote off-colors in sensitive foods and cause lipid oxidation in stored cereals [44]. It has been widely used to fortify liquid milk where it overcomes the inhibitory effects of calcium and casein on iron absorption and has been shown to improve the iron status in children [18]. The cost/mg of iron from ferrous bisglycinate and NaFeEDTA is 13-fold that of ferrous sulphate [16].

#### 2.2.3. Phytic Acid Degradation

Phytic acid in cereal grains can be drastically decreased by the milling of wheat or polishing of rice and can be completely degraded by the addition of the phytase enzyme from *Aspergillus niger* during the manufacture of cereal or soy-based infant foods, or by activation of the native phytases present in wheat or rye [45]. The process, however, involves holding an aqueous slurry of the cereal food for 1–2 h at the optimum temperature and pH for phytase activity [46] followed by a drying procedure. This is an expensive procedure and it is easier and less costly for the manufacturers of cereal-based infant foods to ensure adequate iron absorption by the addition of ascorbic acid.

The addition of phytase to micronutrient powders used for in-home fortification would be far less expensive than phytate degradation during manufacture. Adding phytase at the point of consumption has been reported to degrade phytic acid during the digestion process and to greatly improve iron absorption [47]. The phytase, however, loses its activity above 60 °C and would not withstand addition to cooked cereal porridges that remain above this temperature. The development of a heat-stable phytase would overcome this concern.

## 3. Efficacy Studies

Once the absorption of an adequate amount of iron from a fortified food could be assured, the next step was to demonstrate its efficacy to improve the iron status of women and children. This led to the development of randomized, controlled, double-blind efficacy studies during which children or young women of low or modest iron status consumed the iron-fortified food daily for a period of around 6 months, and iron status was monitored by biomarkers such as serum ferritin (SF)and soluble transferrin receptor (sTfR) in addition to hemoglobin. Gera et al. [48] carried out a meta-analysis of some 60 human efficacy trials with a variety of different iron-fortified foods and concluded that consumption of iron-fortified foods resulted in an improvement in hemoglobin and SF and reduced the risk of remaining anemic and iron deficient. Efficacy studies with iron-fortified staple foods and condiments are further discussed below.

## 4. Fortification Level

Commercial iron-fortified foods usually provide 15–30% of the RDA. Iron-fortified foods which form part of large-scale national programs are more demanding. They aim, with one or more food vehicles, to fill the gap between the current iron intake of the at-risk populations and their iron requirement. The suggested method [17] to define the fortification level for most micronutrients added to staple foods and condiments would be to first measure the current micronutrient intake by the at-risk population groups, and then set the fortification level to increase the micronutrient intake in 97.5% of the most at-risk population group to above their estimated average requirement (EAR), without exceeding the upper limit of the micronutrient in any other population group. However, iron is an exception, as the iron intakes of children and menstruating women are not normally distributed. The WHO [17], therefore, recommends the full probability approach and has published tables for women and children giving the probability of inadequacy at different daily iron intakes in relation to the estimated dietary iron bioavailability (5%, 10%, or 15%). The fortification level is then set to decrease the probability of iron inadequacy to between 2% and 3%.

Unfortunately, many LMICs do not have the necessary dietary data for this methodology, and a more practical approach was used to define the WHO recommendations for the iron fortification of wheat and maize flours [42]. A task force first reviewed published efficacy studies from LMICs, where women and children had consumed a variety of foods fortified with different iron compounds providing varying amounts of iron, and they estimated that a minimum of 7 mg fortification iron per day as ferrous sulfate was needed to usefully improve iron status. This amount, which represents about 50% EAR for both women of child-bearing age and for adolescent boys and girls consuming a 10% iron bioavailability diet, was the basis for the WHO [43] wheat and maize flour fortification recommendations. Estimated average daily flour intake, and the estimated RBV of the other recommended iron fortification compounds, were additionally accounted for.

## 5. An Update of Current Iron Fortification Technologies for Large-Scale Public Health Programs

The technologies for the iron fortification of staple cereals, milk and salt are reviewed in more detail by Hurrell [8].

### 5.1. Wheat and Maize Flours

Both wheat and maize flours have been fortified for several decades with a range of vitamins and minerals. The well-established fortification process involves adding a micronutrient premix precisely onto the milled flour as it moves along a conveyor and then mixing to homogeneity. Wheat and maize flours have established international guidelines for iron fortification [43,49], with ferrous sulfate, ferrous fumarate, NaFeEDTA, and electrolytic iron being the only recommended compounds. These compounds were selected based on sensory studies, efficacy studies with women and children in LMICs, and relative iron absorption in humans. NaFeEDTA is the only recommended compound for high phytate whole grain wheat and maize flours. The evidence to support its use is strong. Iron absorption by human adults from iron-fortified whole-grain wheat bread is reported to be four times higher from NaFeEDTA than from ferrous sulphate [41], and atta flour fortified with NaFeEDTA included in school meals greatly increased the iron status of Indian children [50]. The disadvantage is that the inclusion of NaFeEDTA increases the cost of the micronutrient premix about 3-fold [51].

Efficacy studies with women and children consuming iron-fortified wheat and maize flours were recently reviewed by Hurrell [52] who noted the following caveats. Good efficacy of iron-fortified flours can only be assumed in the absence of widespread inflammation and infections. Only one efficacy study with ferrous fumarate-fortified wheat flour has been published [53], and this did not follow the WHO guidelines with respect to the fortification level and was not efficacious. If ferrous fumarate is to remain on the recommended list, more studies are needed particularly with children, who may absorb iron from ferrous fumarate less well than adults [28]. The recommendations for the iron fortification of maize flour are largely based on studies with wheat flour, and additionally, there are no studies with the high calcium nixtamalized (lime-treated) maize flour used for tortillas.

Not all the efficacy studies with electrolytic iron added to wheat or maize flour improved the iron status of the subjects. The electrolytic iron study that caused the most concern was that of Rohner et al. [12], in which iron-fortified biscuits were fed to 6–14-year-old Ivorian school children for 6 months but resulted in no improvement in iron status. The 9 mg Fe provided daily was almost 2-fold the 50% iron EAR (5.3 mg) for this age group and was expected to be efficacious. High inflammation in the malaria-endemic area of the study could be an explanation, although a more recent study in the same area, in which 1–3-year-old children consumed a maize–soy complementary food fortified with ferrous fumarate and NaFeEDTA for 9 months, reported a significant increase in iron stores [54]. A decreased dissolution of electrolytic iron during digestion in malnourished Ivorian children is a possible explanation.

It is estimated that 161 countries worldwide consume more than the 75 g of flour per capita/d that is needed to adequately fortify cereal flour, and, of these, 83 countries have introduced mandatory legislation for wheat flour fortification [55]. Globally 80% of wheat flour is milled in large industrial mills and 32% is fortified, with the quality of fortification monitored by international standards and subject to government inspections and audits. Relative to other vehicles, wheat flour fortification practices are considered the most advanced, and iron-fortified wheat flour is considered at present to have the highest potential for impact [52]. On the other hand, maize flour fortification is mandatory in only 16 countries, 32% is milled in large industrial mills, of which 54% is fortified. Small industrial maize mills predominate in Africa and Central America where maize is mostly consumed. Such mills often have little process control and make high-quality fortification difficult to ensure.

### 5.2. Rice

Because rice is consumed as a grain and not as a flour, a suitable technology for large-scale rice fortification is not yet in place, and although global rice milling is relatively well consolidated, only 1% of industrially milled rice is fortified [52]. The future, however, looks positive as a new extruded fortified kernel technology has replaced the older coating and dusting technologies which had little success. With the extrusion technology, rice flour is first mixed with a micronutrient premix and water and then extruded to a similar shape and appearance as regular rice kernels, with which they are subsequently blended. Extruded kernels are extremely sensitive to color changes with added iron, and FePP is the only compound that causes no color change [56]. Both hot and cold extrusion techniques have been applied and recently the addition of trisodium citrate and citric acid has been reported as a novel enhancer that will double iron absorption from FePP [57].

Eight efficacy studies in women or children consuming iron-fortified extruded premix rice have been reported, six with micronized ground FePP (MGFP) (particle size < 2.5 μm) and two with micronized dispersible FePP (MDFP) (particle size 0.3 μm and encapsulated). None of the studies included the addition of trisodium citrate and citric acid. Both studies with MDFP reported good efficacy, but the high cost of this compound makes it unsuitable for use in large-scale programs. The efficacy of MGFP-fortified rice depended on the iron fortification level. The three studies that provided an additional iron intake above the 14 mgFe/d recommended for FePP [42] reported good improvements in the iron status in Indian and Brazilian children [58,59,60] whereas two of the three studies providing an additional 7–10 mg of additional Fe reported an inconsistent influence on iron status. One of these studies was made in Cambodia where high inflammation in the child subjects may also be responsible for the low and inconsistent impact of the iron-fortified rice [61].

Rice is consumed mainly in Asia but has also become an important staple in some African and Latin American countries with >50 countries consuming more than the 75 g/capita/d needed for a national iron fortification program. Slightly over 50% of rice for human consumption is industrially milled and could be fortified with micronutrients; however, a large proportion is still milled by farmers in thousands of small- and medium-sized mills using old machinery and technology. In such mills, high-quality fortification would be challenging. Nevertheless, rice fortification is gaining popularity and several countries are introducing iron-fortified rice into public health programs and retail markets [62]. Investment is still required to scale up; the fortification technology may need some minor adjustments and rigorous control and monitoring systems must be put in place.

### 5.3. Salt

Salt is iodized in more than 140 countries worldwide and has had impressive success in eliminating iodine deficiency. It is universally consumed in relatively constant amounts by all population groups independent of socioeconomic status. The question is whether it can be fortified with other micronutrients and have the same public health success in decreasing other micronutrient deficiencies, without jeopardizing the success of the iodine program.

Attempts to develop double-fortified salt (DFS) by adding both iodine and iron have been underway for several decades [63], and although progress has been made, a widely acceptable fortification technology for the large-scale fortification of all salt qualities with iron and iodine still does not exist [52]. There are two main reasons for this. Firstly, there is a wide variation in the quality of raw salt with respect to purity, moisture content, and particle size, and secondly, some iodized salts are extremely sensitive to color changes and iodine losses on addition of iron. Iron can catalyze the oxidation of iodate and iodide during storage to iodine gas and, under unfavorable storage conditions, iodine losses can be almost complete [64]. A third reason is that salt intake is only 5–10 g/d, which can easily carry the μg of iodine needed for fortification but not mg of iron.

Two iron fortification compounds are still under investigation for addition to DFS. Either compound could be easily added to the dry iodized salt in a batch or continuous blender. The simplest, least expensive option would be to add FePP which is a yellowish, white powder. This compound causes few if any color changes in most foods, and there are many reports on its absorption and efficacy. It is widely used to fortify commercial infant cereals and extruded premix rice.

The second option is to add encapsulated ferrous fumarate. This sophisticated capsule has been developed for Nutrition International (NI, formerly Micronutrient Initiative) by the University of Toronto and is still currently undergoing changes. The encapsulated fumarate used in the efficacy studies reported below was manufactured by a granulation process followed by an encapsulation process. Ferrous fumarate, water, hydroxypropyl methylcellulose (HPMC), sodium hexametaphosphate, and titanium dioxide were first agglomerated on a fluidized bed dryer to a similar size as salt grains and then encapsulated with a mixture of titanium dioxide and soy stearine [65]. The current manufacturing process, developed to avoid the appearance of black specks in stored salt, first extrudes ferrous fumarate with cereal flour, water, and vegetable oil, followed by coating with titanium dioxide and encapsulating with HPMC and soy stearine [66]. The titanium dioxide is needed to mask the red color of the ferrous fumarate.

Efficacy studies in Morocco, Cote d’Ivoire, and India, providing 10–18 mg Fe/d, have all reported good improvements in the iron status of school-aged children consuming FePP (MGFP)-fortified DFS. In the Moroccan study [67], IDA decreased from 30% at baseline to 5% after 10 months, and the iodine status improved in a similar way to iodized salt without iron. Similarly, consumption of DFS, providing 12 mg/Fe as encapsulated fumarate, has been reported to improve the iron status of 18–55-year-old women [68] and school-aged children in India [69].

Although good efficacy has been reported for both approaches, further improvements are still needed to prevent adverse color formation and iron-catalyzed iodine losses. In relation to FePP, changing from MGFP (particle size < 2.5 μm) to regular FePP (mean particle size 20 μm) could prevent both the reported yellowing of the salt and the reported iron-catalyzed iodine losses in moist salt. There is no evidence that the larger particle size decreases iron absorption from FePP. If regular FePP still causes unacceptable color changes or iodine losses, then a simple capsule of partially hydrogenated soybean oil, known not to influence iron absorption, could help. Great care must be taken to ensure there are no iron-catalyzed iodine losses so as to avoid jeopardizing the iodization program. DFS production would be straightforward for the large industrial salt producers that account for >70% of industrial salt production for human consumption, but it would be a challenge for the many small-scale producers in coastal areas or along lake shores who produce lower quality salt [70]

More studies are needed to justify the use of NI-encapsulated fumarate. If the current compound is confirmed to cause no color changes or iodine losses and continues to be used, there is a need to investigate whether the sophisticated capsule decreases iron absorption from ferrous fumarate in human studies and to confirm its efficacy in improving the iron status in women or children. Salt is very sensitive to price increases, so the additional cost of using encapsulated fumarate may be an issue. Perhaps more importantly, however, is that the European Food Safety Agency no longer considers titanium dioxide safe when used as a food additive, as they could not exclude its potential to cause genotoxicity [71]. An alternative solution should thus be found to mask the red color of ferrous fumarate.

### 5.4. Milk

The milk industry is well established and has a long history of fortification, especially with vitamins A and D. Milk is also widely consumed worldwide but especially in Latin America, Europe, and Southern sub-Saharan Africa [72]. Reconstituted cow’s milk powder fortified with ferrous sulfate or ferrous gluconate can be a useful vehicle to provide iron to young children, provided that ascorbic acid is added to overcome the inhibition of calcium and casein on iron absorption. The combination of soluble iron with ascorbic acid however causes unacceptable flavor changes to liquid milk and iron fortification has not been widely practiced.

The WHO [17] recommends ferrous bisglycinate (FBG), MDFP, or ferric ammonium citrate for liquid milk fortification. FBG is preferred as it causes no sensory changes and partly overcomes milk’s inhibition of iron absorption. Recently, a novel casein–iron–phosphate complex was developed for liquid milk fortification. It causes no unacceptable sensory changes and iron absorption from whole milk fortified with this compound is comparable to that from whole milk fortified with ferrous sulfate [73]. However, whether its absorption is inhibited by the casein and calcium in milk, remains to be investigated.

There is good evidence that public health interventions providing spray-dried cow’s milk fortified with iron and ascorbic acid improve the iron status in children [74,75], and one efficacy study reported an improved iron status in Saudi children consuming FBG-fortified flavored milk [76]. Costa Rica is the only country worldwide that has mandated the iron fortification of both liquid and powdered milk with FBG and, an effectiveness evaluation of the program, which also includes iron-fortified wheat and maize flour, reported a decrease in anemia prevalence in women and children [77].

## 6. Overcoming the Inflammation Barrier to Efficacious Iron Fortified Foods

In recent years, as more knowledge has accumulated on the nature of iron metabolism, the definition of ID has become more complex. We can now describe two distinct forms of ID. As before, absolute ID covers decreased total body iron, caused by low iron intake, low iron bioavailability, increased iron needs, and/or increased iron losses. The second form of ID has been termed functional ID (FID). It is defined as a decreased iron supply for erythropoiesis, even though the body’s iron stores remain high [78]. FID occurs as a result of inflammation and is the cause of the inflammation barrier to efficacious iron-fortified foods as it cannot be overcome by increasing the iron supply.

As human beings evolved, the physiological effort put into excluding dietary iron from the body was as high as the effort put into acquiring iron for normal body functions [10]. A major part of this effort was the development of an inflammation barrier as a defense mechanism to prevent circulating iron from exacerbating infections. It is therefore unlikely that we could overcome this barrier by dietary manipulation, and it would seem unwise to bypass an integral part of our immune defense system. Ultimately, infections must be controlled so we can remove or greatly lower this barrier but, at present, we need to know at what level of inflammation within an ID population can still permit a useful increase in their iron status. In other words, at what level of inflammation can large-scale iron fortification programs still have a beneficial impact on public health?

Iron absorption and distribution to the tissues are under the strict control of the hormone hepcidin, which balances the need for iron with the potential of iron to cause harm. Hepcidin controls dietary iron absorption by regulating the entry of dietary iron from the duodenal enterocytes into the plasma, and it controls iron distribution to the tissues by regulating the passage of recovered red cell iron from the reticuloendothelial macrophages into the plasma. Hepcidin expression is downregulated by low iron status, increasing iron absorption and increasing iron distribution when the body needs more iron, and hepcidin expression is upregulated by an adequate iron status and high iron stores, decreasing iron absorption and distribution when the need for iron is low. However, hepcidin expression is also upregulated by inflammation, caused by infections, so as to block the pathogen’s access to iron.

It is this infection-related inflammation, in women of reproductive age and in children, that threatens the efficacy of iron-fortified foods and the effectiveness of large-scale iron fortification programs. Iron is an essential micronutrient for nearly all pathogens and decreasing the accessibility of iron to pathogens is an important component of the body’s immune defense. The restricted dietary iron absorption (usually 1–2 mg Fe is absorbed per day) as a result of inflammation can decrease the efficacy of iron-fortified foods and would be expected to decrease the impact of large-scale iron fortification programs. The impaired distribution of recovered red cell iron (usually ca.20 mg Fe is redistributed per day) results in hypoferremia and leads to anemia of inflammation (AI), which cannot be differentiated from IDA caused by low total body iron. AI was previously known as the anemia of chronic disease [78]. The presence of AI thus increases the difficulty of measuring the impact of iron-fortified foods on iron status. In populations living in areas with widespread infections and inflammation, it has been estimated that only 25% and 37%, respectively, of the anemia in preschool children and women of reproductive age, is associated with absolute ID resulting from decreased total body iron (IDA) [79]. Most of the remainder would be expected to be caused by FID due to widespread inflammation (AI).

Petry et al. [79] developed an inflammation exposure score for women and children. The score was based on the prevalence of presumed and confirmed malaria cases; HIV prevalence in adults; obesity prevalence in female adults; schistosomiasis prevalence; and an overall hygiene score based on the proportion of the population using improved drinking water, and the proportion of the population using improved sanitation facilities. The hygiene score was used as a proxy for the risk of enteric inflammation. The inflammation score was applied to 188 countries, which were classified as having low, medium, high, or very high inflammation exposure. All of the 21 countries with very high inflammation exposure, and 22 of the 28 countries in the high inflammation exposure group, were from sub-Saharan Africa. There were 33 countries with medium inflammation exposure, mostly coming from Latin America, the Caribbean, the Middle East, North Africa, and Oceana. Obesity in women was the major factor causing inflammation in many countries in this medium inflammation group, with poor sanitation and unclean water other important factors. In high and very high inflammation countries, obesity in women, poor sanitation, and unclean water also contributed to the inflammation score but it was HIV and malaria that generated the high and very high inflammation exposure scores.

Iron fortification interventions would be expected to be efficacious in countries with low inflammation exposure and it is tempting to suggest that they can also be efficacious in countries with medium inflammation scores. For example, iron efficacy studies with a range of foods and different iron compounds in Morocco, Philippines, Brazil, Mexico, Thailand, and Kuwait, all medium inflammation exposure countries, have reported useful improvements in the iron status of women or children [52]. If this is confirmed, our concern about the impact of inflammation on the efficacy of iron-fortified foods should be focused on countries in sub-Saharan Africa.

Another reason why we are still unclear on whether iron-fortified foods can usefully improve iron status when inflammation exposure is high is that we are also unsure of the true extent of ID in sub-Saharan Africa. The widespread, high prevalence of anemia in women and children in sub-Saharan Africa has led to the belief that absolute ID is also high. However, we have little evidence for this, and it is difficult if not impossible to distinguish between IDA caused by absolute ID and AI caused by FID. Additionally, as well as influencing the form of ID, inflammation also confounds the quantification of ID by having a strong influence on the main iron status biomarkers, SF (or plasma ferritin (PF)) and sTfR, leading to the need for inflammation-corrected measurements [80]. Both forms of ID and anemia are common in clinical practice in patients with diseases such as cancer, heart failure, rheumatoid arthritis, and inflammatory bowel disease, and transferrin saturation and sTfR have been suggested as possible approaches to distinguish between them [78]. Iron status in sub-Saharan African countries with high inflammation exposure has usually been assessed only by anemia prevalence, and only the more recent biomarker studies will have iron biomarkers corrected for inflammation.

The effectiveness of large-scale iron fortification programs has rarely been assessed, even in low inflammation exposure countries, presumably because it was not considered necessary. It was only recently that Costa Rica [77] reported a positive impact of their fortification program on the iron status of women of reproductive age and young children. Barkley et al. [81] reported the first reasonably good evidence that flour fortification in LMICs decreases anemia prevalence in women, by comparing anemia prevalence for countries that had pre- and post-fortification data (*n* = 12) at similar time points with anemia prevalence for countries that never fortified flour (*n* = 20). The 12 food fortification programs investigated included those from six countries with medium inflammation exposure (Bolivia, Fiji, Jordan, Nicaragua, Peru, Philippines) and one (Senegal) with high inflammation. After controlling for the human development index and endemic malaria, the results suggested that anemia prevalence had decreased significantly in countries that fortified flour with iron (and other micronutrients) while remaining unchanged in countries that do not. Each year of fortification was reported to be associated with a 2.4% reduction in the odds of anemia prevalence.

More recently, Engle-Stone [82] measured the effectiveness of a large-scale ferrous fumarate-fortified wheat flour program on the iron status of women and children in Cameroon, a very high inflammation exposure country. Inflammation was reported to be around 20% in women and 40% in children, and malaria was present in some 6% of women and 10% of children. Iron status was monitored by hemoglobin, PF, and sTfR measurements taken 2 years prior to the program and one year after the start. While there did appear to be an improvement in the iron status of both the women and children, the changes were small and inconsistent. Anemia prevalence in women decreased from 47% to 39%, while anemia in children remained unchanged at 47%. ID, based on PF adjusted for inflammation, was 24% at baseline in women and did not significantly decrease, whereas ID in children decreased from 27% to 16%. ID based on sTfR decreased from 31% to 9% in women and from 64% to 25% in children.

These results raise concern over the ability of PF and sTfR to give similar estimates of ID prevalence in the presence of inflammation. They also confirm earlier reports from the Côte d‘Ivoire, that sTfR predicts a much higher prevalence of ID than PF. ID in Ivorian children was estimated to be 40% using corrected sTfR and 4% with corrected PF. The corresponding values in women were 30% vs. 12% [83].

There is better evidence from well-controlled efficacy studies that iron-fortified porridges can improve the iron status of young children living in high inflammation countries. Andang‘o et al. [84] fed NaFeEDTA-fortified maize porridge to 3–8-year-old Kenyan children for 5 months and, in order to avoid the confounding effect of inflammation on iron status biomarkers, gave malaria chemotherapy to infected children prior to the baseline, and to all children prior to the end of the study. Consumption of the iron-fortified porridge decreased anemia from 71–38% and ID, based on non-corrected PF, from 18% to 3%. In a more recent study in the Côte d‘Ivoire, Glinz et al. [54] fed a maize-soy complementary food, fortified with ferrous fumarate and NaFeEDTA, to 1–3-year-old children for 9 months. Throughout the study, some 50% of the children were infected with malaria and around 60% had elevated inflammation biomarkers. Anemia prevalence remained high, at about 80%, in both control and iron-treated children throughout the study. However, at 9 months, ID based on corrected PF had decreased from 40% to 4% in the iron-treated children with no significant change in controls. Clearly, there is a need to improve the measurement of ID in the presence of inflammation and ideally develop the methodology to distinguish absolute ID from FID when both are present in the same individual.

A possible explanation for why iron fortification has little or no effect on the prevalence of anemia in the presence of infection-related inflammation is that when iron supply is severely restricted, the body allows the red cell iron to fall so as to protect the essential iron enzymes in the tissues. This hypothesis by Ganz [85] is based on a report by Bullock et al. [86] that the enzyme aconitase, essential for erythropoiesis, is inactivated by low iron supply, whereas other tissues continue to use Fe. This preferential partitioning of circulating iron to the tissues may also explain the marked differences reported in ID estimates based on sTfR or PF [82,83] as an iron-deficient red cell would continue to increase sTfR levels, while the continued provision of iron to the tissues could increase iron stores and consequently increase PF.

## 7. Safety of Iron Fortified Foods

There is no evidence to indicate that iron-fortified foods are unsafe; however, there is some concern over potential negative health effects in LMICs where malaria is endemic and/or where water quality and sanitation are poor. Iron supplementation and maybe iron from MNPs are reported to exacerbate malarial infection in children. The likely explanation is that the iron influx into the plasma from high-dose oral supplements exceeds the rate of iron binding to transferrin and a quantity of non-transferrin-bound iron (NTBI) is formed. It is proposed that NTBI increases the intensity of malarial infections by increasing the sequestration of malarial-infected red cells in the capillaries of the brain and intestine, causing more cerebral malaria and more bacteremia by increasing the permeability of the intestinal barrier to pathogens [87]. With the exception of MNPs [15], the consumption of iron-fortified foods in malaria-endemic areas has not been found to exacerbate malaria infection, presumably because consumption of iron-fortified foods results in little or no NTBI formation [88]. While the daily provision of micronutrient powders (12.5 mg iron/d) for 6 months to 6–30-month-old Ghanaian children did not increase the incidence of malaria, it did lead to a higher number of hospital admissions from the iron group than from the controls [15].

Much of the fortification iron consumed with an iron-fortified food is not absorbed but makes its way to the stool where it is excreted. In LMICs, when hygiene is poor, the consumption of iron-fortified foods is reported to increase the number of pathogenic bacteria in the stool to the detriment of the beneficial barrier bacteria [89]. While there is little or no evidence that this increases the risk of diarrhea in children consuming iron-fortified foods, there is evidence that oral iron supplements to children in LMICs modestly increase the risk of diarrhea by 11% [90]. The iron dose delivered by MNPs is comparable to that delivered by supplemental iron doses (2 mg iron per kg body weight).

In controlled studies, iron-containing MNPs have been reported to modestly increase respiratory tract infections and the risk for diarrhea in infants; in some cases, the diarrhea is severe and may require hospitalization [13,14]. Long-term consumption of a lower dose of highly available iron (5 mg/d as a mixture of ferrous fumarate and NaFeEDTA) plus the prebiotic galacto-oligosaccharides (GOS), however, mitigated the adverse effects of iron on the gut microbiome and respiratory infection, while modestly improving iron absorption [91] and maintaining a beneficial effect on iron status [92]. An alternative to GOS would be the addition of bovine apo-lactoferrin, as this compound has also been reported to modestly increase iron absorption and mitigate the adverse effects of unabsorbed iron on the infant gut [93].

## 8. The Way Forward

As we move forward, the technical issues needed to optimize the efficacy of iron-fortified foods will be further resolved but, without some action, the inflammation barrier will still stand strong. We urgently need a better understanding of the influence of common infections and inflammation on iron metabolism, and how inflammation impacts the efficacy of iron-fortified foods. We need to understand why sTfR and PF give very different estimates of ID in the presence of inflammation, even when corrected [83]. This is important in obtaining a better estimate of ID in sub-Saharan African countries and other countries with high inflammation exposure. A better understanding of the inflammation barrier is most needed in relation to large-scale iron fortification and biofortification programs and iron-fortified foods for infants and young children distributed by aid agencies. Malarial and gastrointestinal infections are intermittent, and the resulting restriction of iron absorption and iron distribution would also be expected to be intermittent. Obesity in women, however, leads to chronic inflammation and its influence needs quantification, as does the influence of other sources of chronic inflammation.

The influence of inflammation on the efficacy of iron-fortified foods has not been evaluated experimentally. It seems likely that malaria and HIV would be major generators of infection-initiated inflammation, but we cannot compare their importance with gastrointestinal infections or obesity in women. A further problem in assessing ID is our inability to distinguish between the AI caused by FID and IDA due to low total body iron (absolute ID). This adds to the confounding effect of inflammation on iron status biomarkers and makes our current assessments of the need for iron interventions in sub-Saharan Africa, and any assessment of their impact, even more unreliable.

Nevertheless, there are indications that, despite the inflammation barrier, some fortification iron does enter the body and lead to an improvement in iron status. Although the reported improvements in iron status for large-scale programs are relatively small and inconsistent, improved iron status in infants and young children fed iron-fortified foods is more encouraging. Nevertheless, they would certainly be much better without the inflammation barrier. The question is: “Are they good enough in the presence of the inflammation barrier?”

There are no technical innovations that have been developed to safely bypass the inflammation barrier and, at the moment, such innovations seem impossible and unwise. The only way to increase the impact of iron interventions in countries with high inflammation exposure is to lower the inflammation barrier by removing the causes of inflammation. The interventions that are needed in high inflammation exposure countries to improve the efficacy of iron-fortified foods are under the control of governments not food and nutrition scientists. They are likely to include a better control of malaria and HIV infections, installation of clean water supplies and better sanitation to decrease gastrointestinal infections, and programs to decrease the prevalence of obesity in women.

## Data Availability

Not applicable.

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
