# Peer review of "Ensuring the Efficacious Iron Fortification of Foods: A Tale of Two Barriers"

_nutrients, 2022, doi:10.3390/nu14081609_

Round 1

Reviewer 1 Report

Thank you for the opportunity to read this interesting, relevant and thorough review on obstacles and facilitators to improving iron nutrition through fortification.

I have a few comments/questions.

There are trivial grammatical errors throughout the manuscript so will need careful copy editing - eg Banladeshi, aconitrase

The author states in several places that it is not possible to differentiate iron deficiency anaemia from anaemia of inflammation. Although this is true in the most strict definition - looking at the haemoglobin, Hct and indices of a complete blood count does not allow for discrimination (MCV is no longer felt to be a discriminator, hypochromia suggestive but not conclusive) there are iron status markers that are useful even in the setting of inflammation to aid in differentiation - for example discussion, see Camaschella C, Girelli D. The changing landscape of iron deficiency. Mol Aspects Med. 2020.  

Furthermore, I think when the author discusses TfR it would be helpful to specifically mention soluble TfR.

I recognise sTfR is expensive and not yet standardised - however this may improve in the future, and this manuscript looks to the future.

The author states that diets rich in yams would be expected to have more bioavailable iron compared with phytate rich diets - however, yams contain polyphenols that are potent iron absorption inhibitors - are there efficacy studies that have looked at yams, cassava etc and iron absorption? If so it would be useful to have those discussed.

What physiological process could explain why children with ID would not be able to up regulate iron absorption from ferrous fumarate compared with ferrous sulphate? 

Lastly, does the author have insights on the role of micro-milling of grains genetically bred to have higher iron levels (biofortification) and whether this type of processing would improve iron absorption from such crops and hence status.

Author Response

Thank you very much for you comments,  please check the attachment.

Reviewer 2 Report

The paper titled "Ensuring the efficacious iron fortification of foods; a tale of two 2 barriers" deals with the iron fortification of foods, a topic of great interest and relevance for public health. I found the paper very interesting and readable.

Author Response

Thank you very much for your valuable time.